# Correspondence of the Symmetry of Thermodynamic Properties of Matter with the Symmetry of Equations of State

**DOI:** 10.3390/e25111532

**Published:** 2023-11-10

**Authors:** Ti-Wei Xue, Zeng-Yuan Guo

**Affiliations:** Key Laboratory for Thermal Science and Power Engineering of Ministry of Education, Department of Engineering Mechanics, Tsinghua University, Beijing 100084, China; xuetiwei@tsinghua.edu.cn

**Keywords:** thermodynamic property, symmetry, equation of state, ideal gas, ideal dense matter

## Abstract

Thermodynamics contains rich symmetries. These symmetries are usually considered independent of the structure of matter or the thermodynamic state where matter is located and, thus, highly universal. As Callen stated, the connection between the symmetry of fundamental laws and the macroscopic properties of matter is not trivially evident. However, this view is now being challenged. Recently, with symmetry to the ideal gas equation of state (EOS), an ideal dense matter EOS has been proposed, which has been verified to be in good agreement with the thermodynamic properties of high-density substances. This indicates that there is a certain symmetry between the thermodynamic properties of substances in their high- and low-density limits. This paper focuses on the distinctive features and the significance of this symmetry. It is a new class of symmetry that is dependent on the thermodynamic state of matter and can be incorporated into the existing symmetrical theoretical system of thermodynamics. A potential path for developing the EOS theory arising from this symmetry is discussed. EOS at high densities could be developed by correcting or extrapolating the ideal dense matter EOS based on this symmetry, which might fundamentally solve the difficulty of constructing EOS at high densities.

## 1. Introduction

Thermodynamics is a science of symmetry. As early as the 1850s, Clausius [1,2], one of the founders of thermodynamics, attempted to incorporate the first and second laws of thermodynamics into a symmetrical theoretical framework. He regarded the theorem of equivalence between heat and work as the first theorem and the theorem of equivalence between “transformations” as the second theorem and considered them to be of the same kind [3,4]. Following in the footsteps of Clausius, von Oettingen [5] developed a symmetric and complementary framework for thermodynamics. He argued that there is a physical and linguistic symmetry between thermal and mechanical variables and functions [4]. By the early twentieth century, Ammy Noether linked symmetry and conserved quantities and established the well-known Noether’s theorem. She stated that every continuous symmetry of a physical system contains a conservation theorem and vice versa. Against this background, Callen [6,7] introduced the concept of thermodynamic conserved coordinates. Each such conserved coordinate implicates potential physical symmetry. Nevertheless, what we discuss later is not this symmetry in a mathematical sense but symmetry in a more general sense.

Thermodynamics is the study of the constraints imposed by the symmetries of matter on its possible properties [6]. It is generally believed that thermodynamic symmetry is independent of the specific structures or the thermodynamic states of substances and, thus, is highly universal. However, the above view is now being challenged. Recent studies have found that the thermodynamic properties of matter, in themselves, have symmetry [8,9]. By contrast, the thermodynamic properties are directly dependent on the structure of the substance and the thermodynamic state it is in. An increasing number of empirical observations have shown that the thermodynamic properties of matter display simplicity at the two extremities of density and complexity at intermediate densities. As is well known, the ideal gas equation of state (EOS) is a limiting model, and all matter shows this simple regularity in the low-density limit [10]. The form of the ideal gas EOS is as follows:(1)PV=RT,
where *R* is the ideal gas constant. High-density matter without phase transitions also exhibit simple regularities [11,12]. Correspondingly, a general EOS for the high-density limit has been established [8]. It is called the ideal dense matter EOS and has the following form symmetric to the ideal gas EOS:(2)$$TS=R^{\prime }P,$$
*R*′ is the ideal dense matter constant, where *R*′ < 0. It is an entropy-containing EOS, which itself implicitly has a zero point of entropy. With this zero point chosen, the entropy of the thermodynamic state is always negative. The ideal gas and the ideal dense matter are symmetric thermodynamic concepts [8]. An ideal gas is a low-density limiting state with high temperature and low pressure, while ideal dense matter is a high-density limiting state with low temperature and high pressure. It should be emphasized that the model of ideal dense matter does not contradict the third law of thermodynamics (the Nernst heat theorem) because they address different thermodynamic states of matter. In addition, ideal dense matter is different from the concept of “ideal solid” in the field of high pressure [13]. The latter was defined as having the same vibrational circular frequency in lattice dynamics. As with ideal gas, ideal dense matter is hypothetical and practically non-existent. Substances at high densities in agreement with Equation (2) can be regarded as ideal dense matter. Property data have verified that the ideal dense matter EOS has a higher descriptive accuracy for substances with higher densities (higher pressures or lower temperatures) [8]. Since EOS expresses the thermodynamic properties of matter, the symmetry of EOS corresponds to the symmetry between the thermodynamic properties of high- and low-density substances. The symmetry in thermodynamic properties of matter enriches the contents of thermodynamic symmetry, as well as shocking or changing some of the traditional perceptions about thermodynamic symmetry to some extent, which might lead to a new research paradigm for the EOS theory.

## 2. The Ideal Dense Matter EOS and Symmetry of Thermodynamic Properties of Matter

### 2.1. The Ideal Dense Matter EOS from Thermodynamic Symmetry

In von Oettingen’s symmetric and complementary framework, for each pair of relations, a pair of symmetric ones can arise from symmetric transformation expressed as the exchange of variables, *T* ↔ *P* and *S* ↔ −*V* [4,5]. From a typographical point of view, this complementarity leads to two columns of exposition. The Thermodynamic Wheel of Connections (TWC) [14] can be used to express this symmetric framework. As shown in Figure 1, there are two undervalued physical quantities, *C_S_* and *C_T_*, which are symmetrical to specific heat (capacity) at a constant volume, *C_V_*, and specific heat at constant pressure, *C_P_*, respectively. *C_S_* and *C_T_* are called the specific work (capacity) at constant entropy and the specific work at constant temperature. *C_V_* and *C_P_* are defined as follows: (3)$$C_{V}={\left(\frac{\partial U}{\partial T}\right)}_{V}=T{\left(\frac{\partial S}{\partial T}\right)}_{V};\;C_{P}={\left(\frac{\partial H}{\partial T}\right)}_{P}=T{\left(\frac{\partial S}{\partial T}\right)}_{P}.$$
Then, the definitions of *C_S_* and *C_T_* can be obtained directly from the definitions of *C_V_* and *C_P_* using the exchange of variables *T* ↔ *P* and *S* ↔ −*V*.
(4)$$C_{S}={\left(\frac{\partial U}{\partial P}\right)}_{S}=-P{\left(\frac{\partial V}{\partial P}\right)}_{S};\;C_{T}={\left(\frac{\partial F}{\partial P}\right)}_{T}=-P{\left(\frac{\partial V}{\partial P}\right)}_{T}.$$
*C_S_* and *C_T_* play an important role in the ideal dense matter EOS.

Based on the ideal gas EOS, the expression for the ideal dense matter EOS, Equation (2) was obtained immediately with the exchange of variables *T* ↔ *P* and *S* ↔ −*V*. As with the ideal gas EOS, the ideal dense matter EOS has a very simple form. Thermodynamic symmetry implies that the ideal dense matter EOS should have good agreement with matter at high densities. A comparison with actual property data of various substances has shown that the higher the density of the matter, the more accurate the description of the ideal dense matter EOS is, as shown in Figure 2. It has also been proved to be a reduction of some empirical EOS such as the Tait equation and the Kumar equation in the high-density limit [8].

It follows from the forms of the ideal gas EOS and the ideal dense matter EOS that it is simpler to describe the thermodynamic properties of low-density substances using volume-containing EOS; by contrast, it is simpler to describe the thermodynamic properties of high-density substances using entropy-containing EOS. Temperature plays a more important role in the thermodynamic properties of low-density substances than pressure, so temperature appears alone on one side of the ideal gas EOS. Pressure plays an important role in the thermodynamic properties of high-density substances than temperature, so pressure appears alone on one side of the ideal dense matter EOS.

### 2.2. Symmetry of Thermodynamic Properties of Matter

According to the ideal dense matter EOS, both the internal energy and the Helmholtz free energy in the high-density limit depend on pressure only [8]:(5)$$dU=C_{S}dP;\;dF=C_{T}dP,$$
which is symmetrical with the characteristics of matter in the low-density limit where both internal energy and enthalpy depend on temperature only, as shown:(6)$$dU=C_{V}dT;\;dH=C_{P}dT.$$
Furthermore, for the high-density limit, the following parametric relation can be derived:(7)$$C_{T}=C_{S}-R^{\prime },$$
which is symmetrical with Mayer’s formula in the low-density limit:(8)$$C_{P}=C_{V}+R.$$
Moreover, assuming that *C_S_* or *C_T_* is constant, the expression for the volume in the high-density limit is as follows [8]:(9)$$V=-C_{T}\ln\frac{P}{P_{0}}-R^{\prime }\ln\frac{T}{T_{0}}+V_{0},$$
which is symmetrical with the expression for entropy in the low-density limit:(10)$$S=C_{P}\ln\frac{T}{T_{0}}-R\ln\frac{P}{P_{0}}+S_{0},$$
where the subscript 0 denotes a thermodynamic reference state. The symmetry between the thermodynamic properties of the high- and low-density limits leads to the symmetry between the expressions of individual thermodynamic functions or variables in these thermodynamic limiting states. 

The thermodynamic functions and mathematical relations, such as Maxwell relations, are intrinsic in the macroscopic theoretical system of thermodynamics. Their definitions are independent of the specific structures or the thermodynamic states of substances. The symmetries between them are reasonable as long as they exist. However, the thermodynamic properties of specific substances, in themselves, are not universal and need refinement via the inclusion of an atomistic picture [15]. As Callen [6] stated, the connection between the symmetry of fundamental laws and the macroscopic properties of matter is not trivially evident. The attempt to derive the latter from the former has never been seen before. It is generally considered that problems may arise when symmetry is applied to explore the macroscopic properties of substances. The establishment of the ideal dense matter EOS broke this taboo and successfully linked symmetry with the thermodynamic properties of matter. This means that not only the definitions of thermodynamic functions can have symmetry but also that their expressions for specific thermodynamic states can have symmetry. The former is universal, the latter is dependent on thermodynamic states-. The latter point is one not explicitly aware of before. The symmetry in the thermodynamic properties of matter, accompanied by the symmetry between the ideal gas EOS and the ideal dense matter EOS, is a well-coordinated supplement to von Oettingen’s dual framework.

## 3. Developments of EOS Theory Based on Symmetry of Thermodynamic Properties of Matter

The discovery of symmetry in the thermodynamic properties of matter not only brings cognitive improvement but also opens up more possibilities for the development of the EOS theory. 

The ideal gas EOS is the low-density limit model and has generality. It deals only with commonalities in thermodynamic behavior. The extrapolation based on the ideal gas EOS is a popular method to construct an EOS for actual matter [16,17,18]. However, for high-density matter far from the ideal gas state, extrapolation becomes difficult due to complicated and diverse molecular interactions. Instead, a great number of empirical EOS have been established [19,20,21,22,23,24]. They usually have complicated forms or a large number of coefficients. Most of these coefficients are “isolated”, and their values can only be identified using a large amount of experimental data [25,26]. When these empirical EOS are further extrapolated beyond their range fitted to experimental data, the results are commonly unreliable [27,28].

The above problem can hopefully be resolved based on the symmetry between the thermodynamic properties of high- and low-density substances. According to thermodynamic symmetry, the ideal dense matter EOS is a high-density limit model and has generality like the ideal gas EOS. Studies have shown that the higher the density, the higher the description accuracy of the ideal dense matter EOS [8]. Therefore, the EOS at high densities can be constructed using the ideal dense matter EOS, instead of the ideal gas EOS, as a basic framework. The distinctive feature of this approach is that the higher the density of matter, the fewer modifications are taken, but the higher the accuracy is instead.

The Tait equation is a well-known empirical equation describing high-pressure substances with the following expression [29]:(11)$$V=-D\ln\frac{P+E}{P_{0}+E}+V_{0},$$
where *D* and *E* are the isothermal coefficients. According to Equation (9), the isothermal *P*-*V* EOS of ideal dense matter is the following:(12)$$V=-C_{T}\ln\frac{P}{P_{0}}+V_{0}.$$
Immediately, the Tait equation can be viewed as a modification of the ideal dense matter EOS. As the pressure tends toward infinity, the Tait equation reduces to the ideal dense matter EOS. By comparison, the physical meaning of the empirical coefficient, *D*, is found. It is just the specific work at a constant temperature, *C_T_*. 

In addition, there is a general EOS proposed by Kumar for solids at high pressures [30]:(13)$$P=\frac{B}{A}\left[\exp A\left(1-\frac{V}{V_{0}}\right)-1\right],$$
where *A* and *B* are constants. Compare it with the isothermal *P*-*V* EOS of the ideal dense matter under pressure representation:(14)$$P=P_{0}\exp\frac{V_{0}}{C_{T}}\left(1-\frac{V}{V_{0}}\right).$$
Similarly, the Kumar equation can be seen as a modification of the ideal dense matter EOS under pressure representation.

Actual substances exhibit an ideal gas behavior toward the low-density limit and an ideal dense matter behavior toward the high-density limit. The thermodynamic state at intermediate densities probably has some features of both these limits and is likely to be some kind of superposition of them. Therefore, for intermediate-density substances that are far from both the ideal gas limit and the ideal dense matter limit, their EOS can be constructed using both the ideal gas EOS and the ideal dense matter EOS as basic frameworks. Recently, a global EOS for arbitrary densities has been developed by interpolating between the ideal gas EOS and the ideal dense matter EOS [9]. Since information from both is included, the global EOS has been verified to have a higher descriptive accuracy over a wide range of physical properties with fewer coefficients compared to other empirical EOSs, as shown in Figure 3. It was also demonstrated to be highly predictive in the field of explosive physics and shock compression [9].

In fact, such an equation containing information regarding these two limits already existed before the ideal dense matter EOS was established. For example, the Jones–Wilkins–Lee (JWL) EOS is quite effective EOS in describing the expansion behavior of explosion products [31,32]. Its simplified form is as follows:(15)$$P=ae^{-R_{1}V}+\frac{RT}{V},$$
where *a* and *R*_1_ are constants. The first term on the right side of Equation (15) denotes the high-pressure contribution, and the second term denotes the low-pressure contribution. The former is essentially the isothermal form of the ideal dense matter EOS, Equation (9), while the latter is exactly the ideal gas EOS. 

## 4. Conclusions

Recently, an ideal dense matter EOS symmetric to the ideal gas EOS was proposed, indicating that there is a certain symmetry between the thermodynamic properties of substances in high- and low-density limits. It is generally believed that thermodynamic symmetry is highly universal and independent of the structure of matter or the thermodynamic state where matter is located. The discovery of symmetry in the thermodynamic properties of matter challenges this perception of thermodynamic symmetry to some extent. Not only are there symmetries in thermodynamic state functions as well as thermodynamic relations, but there are also symmetries in the thermodynamic properties of matter. The former is universal; the latter is state-dependent. The symmetry in the thermodynamic properties of matter is a new class of symmetry and is a coherent supplement to the symmetrical framework of thermodynamics.

The discovery of the symmetry between the thermodynamic properties of high- and low-density substances is of great significance for the development of the EOS theory. According to this symmetry, an ideal dense matter EOS for the high-density limit was already established. A common approach for developing an EOS at low densities is to make corrections or extrapolations to the ideal gas EOS. Similarly, EOS at high densities can be developed by correcting or extrapolating the ideal dense matter EOS. Furthermore, EOS at intermediate densities can be constructed using both the ideal gas EOS and the ideal dense matter EOS as the basic frameworks. This might become a new research paradigm of the EOS theory.

## Figures and Tables

**Figure 1 entropy-25-01532-f001:**
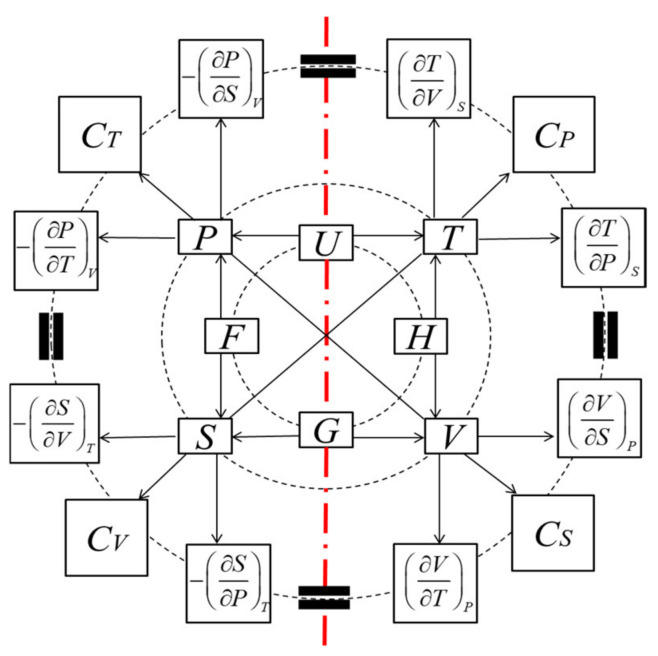
Thermodynamic Wheel of Connections (TWC) [14]. The direction of arrows suggests the partial derivation.

**Figure 2 entropy-25-01532-f002:**
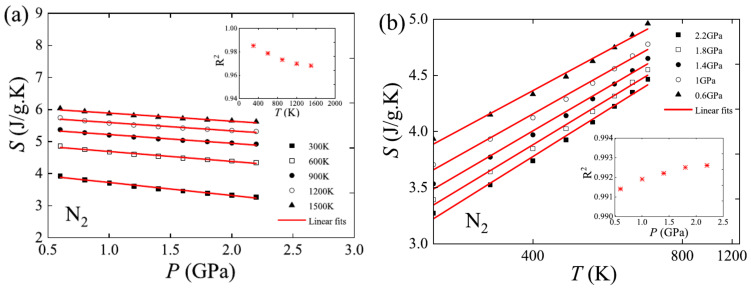
The verification of the ideal dense matter EOS using N_2_ data [8]. (**a**) Isotherms, entropy versus pressure; (**b**) Isobars, entropy versus reciprocal temperature. The insets show the fitting degree. Data are from the National Institute of Standards and Technology (NIST) database. The zero point of entropy for these data is the normal boiling point (NBP), which is different from the implied zero point in Equation (2), so a constant is introduced into the term of entropy in Equation (2) to ensure Equation (2) have the zero point of NBP so that it can be verified.

**Figure 3 entropy-25-01532-f003:**
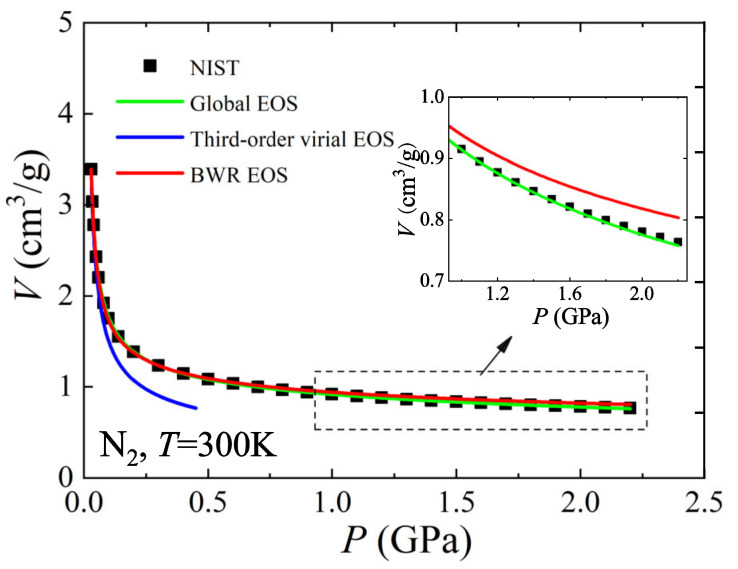
Comparison of the global EOS with the third-order virial EOS and the BWR EOS using N_2_ data [9].

## Data Availability

Data are contained within the article.

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
