# Peer review of "Correspondence of the Symmetry of Thermodynamic Properties of Matter with the Symmetry of Equations of State"

_entropy, 2023, doi:10.3390/e25111532_

Round 1
Reviewer 1 Report
Comments and Suggestions for Authors
In this article, the authors discussed the symmetry between the thermodynamic properties of matter in the high- and low-density limits, and considered the potential for the development of equation of state theory arising from this symmetry. However, it should be noted that this theory has already been proposed in previous literature [Xue T. W.; Guo Z. Y. A general equation of state for high density matter from thermodynamic symmetry. J. Appl. Phys. 2022, 131(4), 044902.]. Therefore, considering the repetition rate of research papers, hoping author make the following improvements to the article:
1. The article only discusses the symmetry between the thermodynamic properties of matter in the high- and low-density limits and mentions the potential of state equation theory development caused by this symmetry, but does not use it specifically to further prove this result. If possible, the author can verify the accuracy of the mentioned theory by comparing it with existing research results.
2. Since the relevant theory has been studied in the literature [1], considering the repetition rate of the research article. The author can explore new aspects or expand the existing knowledge to provide a more comprehensive and insightful analysis.
3. To improve the readability of the article, the author can extract the highlights of the article.
Reviewer 2 Report
Comments and Suggestions for Authors
Round 2
Reviewer 1 Report
Comments and Suggestions for Authors
I think this paper is well revised and can be accepted for publication.
Author Response
Thanks for the reviewer's approval.
Reviewer 2 Report
Comments and Suggestions for Authors
The changes made in response to the reviewers comments are completely adequate. I am looking forward to seeing the paper published.
Author Response
Thanks for the reviewer's approval.